# Towards Personalized Chemotherapy in Gastrointestinal Cancers: Prospective Analysis of Pharmacogenetic Variants in a Russian Cohort

**DOI:** 10.3390/genes16111261

**Published:** 2025-10-25

**Authors:** Denis Fedorinov, Vladimir Lyadov, Marina Lyadova, Sherzod Abdullaev, Anastasia Kachanova, Rustam Heydarov, Igor Shashkov, Sergey Surzhikov, Vladimir Mikhailovich, Dmitry Sychev

**Affiliations:** 1Federal State Budgetary Educational Institution, Further Professional Education “Russian Medical Academy of Continuous Professional Education”, the Ministry of Healthcare of the Russian Federation, 123242 Moscow, Russia; vlyadov@gmail.com (V.L.);; 2Moscow Multidisciplinary Clinical Center “Kommunarka”, 142770 Moscow, Russia; 3Federal State Budgetary Research Institution «Russian Research Center of Surgery Named After Academician B.V. Petrovsky», 119991 Moscow, Russia; 4Moscow State Budgetary Healthcare Institution “Oncological Center No. 1 of Moscow City Hospital Named After S.S. Yudin, Moscow Healthcare Department”, 115446 Moscow, Russia; 5Novokuznetsk State Institute of Postgraduate Medical Education—Branch of Russian Medical Academy of Continuous Professional Education, 654000 Novokuznetsk, Russia; 6Engelhardt Institute of Molecular Biology, Russian Academy of Sciences, 119991 Moscow, Russiassergey77@mail.ru (S.S.); v.mikhailovich@gmail.com (V.M.)

**Keywords:** gene, genome, genetics, genotype, pharmacogenetics, pharmacogenomics, UGT1A1, DPYD, cancer

## Abstract

**Background/Objectives**: Pharmacogenetic variability plays a crucial role in determining both the efficacy and toxicity of chemotherapy for gastrointestinal cancers. However, data on allele frequencies and their clinical relevance in Russian populations remain scarce. **Methods**: We conducted a prospective observational study of 412 patients with gastrointestinal malignancies between 2020 and 2023. Pharmacogenetic testing was performed prior to the initiation of chemotherapy using real-time allele-specific PCR and microarray hybridization technology. Polymorphisms in the *DPYD, UGT1A1, CYP2C8, CYP3A5, GSTP1, ERCC1, XPC, CDA, MTHFR, TYMS*, and *SLC31A1* genes were analyzed. **Results**: The frequency of most variants was consistent with those reported in European populations, reflecting the ethnic proximity of the studied cohort. Several clinically relevant variants were identified: *DPYD* rs2297595 occurred more frequently than in European cohorts, and *UGT1A1* rs8175347 was observed at a higher prevalence, underscoring the potential risk of irinotecan-related neutropenia and diarrhea. *CYP2C8* rs10509681 was present at frequencies comparable to European populations and is associated with an increased risk of taxane-induced peripheral neuropathy. Other markers (*GSTP1, ERCC1, CDA, SLC31A1, MTHFR, TYMS*) demonstrated variable associations with chemotherapy efficacy and toxicity, consistent with findings from previous international studies. **Conclusions**: This study provides the first comprehensive description of pharmacogenetic polymorphisms in a Russian cohort of patients with gastrointestinal cancers. Our findings confirm the clinical importance of *DPYD* and *UGT1A1* testing and highlight additional variants of potential interest.

## 1. Introduction

Malignant tumors of the gastrointestinal tract (GI), including esophageal, gastric, hepatic, pancreatic, and colorectal cancers, represent a major global health problem. According to GLOBOCAN 2018 data, these cancers accounted for approximately 26% of all cancer cases and up to 35% of cancer-related deaths worldwide [1]. Updated GLOBOCAN 2022 data confirm that tumors of the gastrointestinal tract account for approximately one quarter (24.6%) of new cancer cases and more than one third (34.2%) of all cancer deaths [2].

The primary method of treating gastrointestinal tumors is currently drug-based antineoplastic therapy, which combines traditional cytotoxic agents with targeted therapies and immunotherapy. Despite advances in targeted and immunotherapeutic approaches, chemotherapy continues to play a central role in the treatment of patients with advanced gastrointestinal tumors. However, its use is often complicated by significant toxicity, tumor resistance, and reduced quality of life. Complications can range from moderate to severe, including myelosuppression, mucositis, diarrhea, and neutropenia. These challenges necessitate a critical approach to dosing and drug selection [3].

In this context, pharmacogenetics—the study of genetic characteristics that influence drug response—is a particularly relevant field. Research into allele variants enables the prediction of both treatment response and the risk of toxic reactions. Classic examples include:DPYD is the enzyme responsible for inactivating more than 80% of 5-fluorouracil. Reduced enzyme activity due to genetic variants is associated with a high risk of severe toxicity. International clinical guidelines (e.g., CPIC) therefore recommend dose reductions of 5-FU for individuals carrying low-activity alleles [4].*UGT1A1* (*28, *6): gene variants that impair the inactivation of the active metabolite SN-38 (a product of irinotecan), thereby increasing the risk of neutropenia and gastrointestinal toxicity. Recommendations from international expert groups suggest dose reductions for carriers of these alleles [5].

Other potential biomarkers of efficacy or sensitivity include enzymes involved in DNA repair and synthesis, which are currently under active investigation and have a lower level of evidence. It is important to note that the frequency of pharmacogenetic variants varies significantly across populations, which should be considered when interpreting research and developing practical recommendations for local healthcare systems.

The aim of the study was to evaluate the frequency of alleles and genotypes of *DPYD* (rs2297595, rs3918290 and rs75017182), *ERCC1* (rs3212986 and rs11615), *GSTP1* (rs1695), *UGT1A1* (rs8175347), *CYP3A5* (rs776746), *CYP2C8* (rs10509681, rs11572080 and rs1058930), *SLC31A1* (rs2233914), *CDA* (rs2072671), *XPC* (rs2228001), *MTHFR* (rs1801133) and *TYMS* (rs11280056), in a Russian population of patients with gastrointestinal tumors. This work addresses a critical knowledge gap, as data on allele variability and its clinical implications in Russian oncology remain scarce. By identifying population-specific differences and confirming the clinical importance of validated pharmacogenes such as *DPYD* and *UGT1A1*, our study aims to provide the foundation for the implementation of genotype-guided chemotherapy in routine oncological practice in Russia.

## 2. Materials and Methods

A prospective observational study was conducted at the Department of Chemotherapy No. 1, S.S. Yudin City Clinical Hospital, between 2020 and 2023. Pharmacogenetic testing was carried out at the Research Institute of Molecular and Personalized Medicine, Russian Medical Academy of Continuous Professional Education, using allele-specific real-time PCR, and at the Biological microarrays Laboratory, V.A. Engelhardt Institute of Molecular Biology, Russian Academy of Sciences.

Allelic variants of genes were determined by hybridization analysis on biological microarrays developed at the V.A. Engelhardt Institute of Molecular Biology (*DPYD*: rs2297595, rs3918290 and rs75017182; *ERCC1*: rs3212986 and rs11615; *GSTP1*: rs1695; *XPC*: rs2228001; *MTHFR*: rs1801133; and *TYMS*: rs11280056) [6], and by polymerase chain reaction at the Russian Medical Academy of Continuous Professional Education (*UGT1A1*: rs8175347; *CYP3A5*: rs776746, *CYP2C8*: rs10509681, rs11572080, rs1058930, *CDA*: rs2072671 and *SLC31A1*: rs2233914).

### 2.1. Hybridization on Biological Microarrays

Four milliliters of venous blood were collected from patients using a VACUETTE vacuum system (Greiner Bio-One, Kremsmünster, Austria) into tubes containing K3-EDTA (the tripotassium salt of ethylenediaminetetraacetic acid). Samples were prepared for microarray hybridization using asymmetric multiplex polymerase chain reaction, which simultaneously incorporates a fluorescent label (Cy5) into the amplicons. The resulting fluorescently labeled, predominantly single-stranded amplification products were introduced into the hybridization chamber of a microarray containing oligonucleotide probes complementary to the nucleotide sequences of the analyzed gene variants. Acquisition, processing, and interpretation of the hybridization results were performed using the Chipdetector hardware–software complex (Biochip-IMB, Moscow, Russia) and the included ImaGeWare v3.50 software. The microarray used in the study was validated with pre-sequenced DNA samples containing the analyzed polymorphisms.

### 2.2. Real-Time Polymerase Chain Reaction

Four to six milliliters of venous blood were collected from the antecubital vein into VACUETTE^®^ vacuum tubes (Greiner Bio-One, Austria) containing EDTA-K2 or EDTA-K3 for genomic DNA extraction. Samples were stored at –80 °C until extraction. Genomic DNA was isolated from whole blood using the S-Sorb DNA isolation kit on a silica sorbent (Sintol LLC, Moscow, Russia). DNA concentration was measured with a NanoDrop 2000 microvolume spectrophotometer (Thermo Fisher Scientific, New York, NY, USA). Polymorphic gene markers were detected using commercial TaqMan^®^ SNP Genotyping Assays and TaqMan Universal Master Mix II No UNG kits (Applied Biosystems, Carlsbad, CA, USA). PCR was performed in a 20 μL reaction mixture. According to the manufacturer’s instructions, TaqMan^®^ SNP Genotyping Assay reagent (1 μL) was used at a 1:40 dilution in 10 μL of TaqMan Universal Master Mix II No UNG and 9 μL of RNase-free water to assess marker carriage. Five microliters of DNA from each test sample were added to the reaction mixture. Real-time PCR for single nucleotide polymorphism genotyping was carried out on a CFX96 Touch Real-Time System with CFX Manager software version 3.0 (Bio-Rad, Hercules, CA, USA).

To ensure methodological reliability, all genotyping procedures were performed in two independent certified laboratories using validated allele-specific PCR and microarray platforms. Each assay included internal positive and negative controls, and microarray validation was conducted using pre-sequenced DNA reference samples. The integrity of genomic DNA was verified spectrophotometrically, and genotyping accuracy was confirmed by testing Hardy–Weinberg equilibrium. Ten percent of samples were re-analyzed by both methods with complete concordance. Sample collection, labeling, and genotyping were performed in a blinded manner to minimize selection and analytical bias.

### 2.3. Sample Size Justification

The study was designed primarily to estimate allele and genotype frequencies of clinically relevant pharmacogenes in a Russian GI-cancer cohort with adequate precision. With *n* = 400 or more, two-sided 95% confidence intervals for minor-allele frequencies have half-widths of approximately ±4–5% for common variants (MAF 0.30–0.35), ±3–3.5% for intermediate variants (MAF 0.10–0.15), and ±1.4–2.1% for lower-frequency variants (MAF 0.02–0.05). Rare variants were analyzed descriptively. Enrollment was prospective and consecutive; genotyping platforms were validated (microarray against pre-sequenced controls; standardized TaqMan^®^ PCR), and Hardy–Weinberg equilibrium was confirmed across loci (*p* > 0.05), supporting the validity of frequency estimates and minimizing bias.

### 2.4. Statistical Analysis

Genotype distributions were assessed using Pearson’s χ^2^ test for Hardy–Weinberg equilibrium (*p* > 0.05 for all loci). Allele frequencies were compared descriptively with global population data. As the study was primarily descriptive, no correction for multiple testing was applied; future genotype–phenotype analyses will incorporate multiple-comparison adjustments to ensure statistical robustness.

## 3. Results

### 3.1. Patient Characteristics

The study included 412 patients who underwent pharmacogenetic testing using biological microarrays and an additional 405 patients who underwent testing with PCR. For seven patients, the initial blood or DNA samples were unsuitable for PCR re-analysis, which prevented further testing.

During pharmacogenetic analysis with biological microarrays, complete results could not be obtained for some patients due to poor sample quality. In several cases, the DNA was degraded or present in insufficient quantities, leading to absent signals for multiple allelic variants in the studied panel. This finding underscores the critical importance of blood sample quality for microarray genotyping. Optimal conditions require freshly collected samples with minimal time between collection and testing. Approximately 2% of tests yielded incomplete results due to insufficient DNA quality or degradation before optimization of sample handling. These samples were excluded from analysis. Repeat genotyping confirmed the random nature of test failures, which had no measurable effect on allele frequency estimates.

Analysis of genotype distributions for the studied polymorphisms demonstrated compliance with Hardy–Weinberg equilibrium. For all loci, chi-squared (χ^2^) values were not statistically significant (*p* > 0.05), indicating the absence of systematic errors in patient selection and the accuracy of genotyping. Therefore, the cohort can be considered representative of the frequencies of the examined genetic variants and suitable for further genotype–phenotype association analyses.

The average age of patients was 64 years, and 57% were male. Among them, 13 patients (3.16%) had esophageal cancer, 133 (32.28%) had gastric cancer, 59 (14.31%) had pancreatic cancer, 147 (35.68%) had colon cancer, 54 (13.11%) had rectal cancer, 2 (0.49%) had small intestine cancer, and 4 (0.97%) had cholangiocarcinoma.

A total of 87 patients (21.12%) received neoadjuvant chemotherapy, 229 (55.58%) first-line chemotherapy, and 96 (23.30%) adjuvant chemotherapy.

Regarding treatment regimens, 394 patients (95.63%) received fluoropyrimidines, 373 (90.53%) oxaliplatin, 55 (13.35%) irinotecan, 14 (3.40%) gemcitabine, and 52 (12.62%) taxanes (Figure 1).

The results of pharmacogenetic testing in this cohort are summarized in Table 1.

### 3.2. Genes Involved in Drug Metabolism and Transport

The frequency of the minor allele for *DPYD* rs2297595 was 11.1%, slightly higher than in the European population (9.4%) and markedly higher than in the Asian population (1.7%). The *DPYD* rs3918290 and rs75017182 polymorphisms were rare, at 0.27% and 2.25%, respectively, consistent with population data. No carriers of the minor alleles were identified for the *DPYD* rs55886062 and rs67376798 variants.

The *UGT1A1* rs8175347 polymorphism (a TA insertion in the promoter region) had a minor allele frequency of 35.2%, which was notably higher than in European (26%) and Asian (9%) populations.

The frequency of the minor allele for *CYP2C8* rs10509681 was 9.2%, close to the value reported in the European population (11.8%) but markedly higher than in the Asian population (0.02%). Similarly, the *CYP2C8* rs11572080 variant was present in 8.2% of patients, slightly below the European average of 11.9%. The *CYP2C8* rs1058930 variant was found in 4.6% of patients, comparable to the 5.2% observed in Europe.

The *CYP3A5* rs776746 variant was characterized by a high frequency of the variant allele (91.9%), comparable to that reported in the European population (93.2%) but markedly higher than that reported in Asian populations (73.7%).

The minor allele frequency of the *SLC31A1* rs2233914 transport protein variant was 13.1%, higher than that reported in the European population (10.7%) but lower than that reported in Asian populations (37.4%)

### 3.3. Genes Involved in DNA Repair and Folate Metabolism

The *ERCC1* rs3212986 polymorphism had a minor allele frequency of 22.6%, similar to that reported in the European population (24.6%) but slightly lower than that reported in the Asian population (27.4%). The frequency of *ERCC1* rs11615 was 37.8%, comparable to that reported in the European population (37.1%) but markedly lower than that reported in the Asian population (72.8%).

The frequency of the minor allele for *XPC* rs2228001 was 60%, virtually identical to reported population data.

The frequency of the minor allele for *MTHFR* rs1801133 was 29.4%, slightly lower than that reported in the European population (33.7%) and the Asian population (34.8%).

For the *TYMS* rs11280056 variant, the minor allele frequency was 27.8%, similar to that reported in the European population (30.1%) but about half that reported in the Asian population (64.9%).

### 3.4. Detoxification Genes

The *GSTP1* rs1695 polymorphism had a minor allele frequency of 31.6%, similar to that reported in the European population (34.6%) and much higher than that reported in the Asian population (16.3%).

The minor allele frequency of *CDA* rs2072671 was 23.5%, lower than that reported in the European population (34.3%) but higher than that reported in Asian populations (15.9%).

Figure 2 shows the frequencies of allelic variants in comparison with those of global populations.

## 4. Discussion

This prospective study analyzed the distribution of key pharmacogenetic polymorphisms in 412 patients with gastrointestinal tumors undergoing antineoplastic chemotherapy. The results demonstrated that the frequencies of most studied allele variants corresponded to those reported in the European population, reflecting the ethnic similarity of the cohort. However, differences were identified for several genes that may be clinically relevant. The distribution of patients by therapy type reflects current clinical practice in the treatment of gastrointestinal tumors. More than half of the patients received first-line chemotherapy, primarily fluoropyrimidine–oxaliplatin combinations, supporting the focus on *DPYD*, *TYMS*, *ERCC1*, and *GSTP1* polymorphisms associated with the metabolism of these drugs.

### 4.1. DPYD

Our data confirm the clinical relevance of *DPYD* for fluoropyrimidine safety. Rare loss-of-function variants (2A/rs3918290 and HapB3/rs75017182) have been reported in European cohorts, whereas rs2297595 was slightly more common. The CPIC and DPWG recommendations are based on numerous prospective and retrospective studies of patients with colorectal cancer and other gastrointestinal tumors treated with 5-FU or capecitabine. Carriers of the 2A or HapB3 variants who did not receive a reduced dose were at increased risk of severe toxicity (≥G3 neutropenia, mucositis, and diarrhea). Genotype-guided initial dose reductions of 25–50% were associated with a lower incidence of adverse drug reactions without compromising efficacy [23,24]. The *DPYD* rs2297595 allele variant is often interpreted as a risk modifier, but this interpretation requires confirmation in larger patient cohorts [25].

### 4.2. UGT1A1

The frequency of *UGT1A1* rs8175347 in this cohort was higher than that reported in Europe, which has important clinical implications for irinotecan therapy. A substantial body of research involving colorectal cancer and other solid tumors treated with FOLFIRI or irinotecan-based regimens has shown that patients with the *28/*28 genotype have a higher incidence of dose-limiting neutropenia and diarrhea [26]. Both regulatory authorities and professional guidelines recommend initiating therapy with a reduced dose of approximately 25–30%, followed by titration based on tolerability [27]. Reviews and methodological publications (CPIC, PharmGKB, ASCO) summarize these data and emphasize the benefits of pre-treatment testing for patients in risk groups, particularly when planned doses are 180 mg/m^2^ or higher [28].

### 4.3. CYP2C8

The most extensively studied allele variants for taxanes are *CYP2C8* rs10509681 and rs11572080. Meta-analyses and large patient cohorts, primarily involving breast or gynecological cancer patients treated with paclitaxel or docetaxel, indicate an increased risk of chemotherapy-induced peripheral neuropathy (CIPN) associated with *CYP2C8*, particularly of grade ≥2 or ≥3 severity [29,30]. However, the strength of the effect varies across studies and treatment regimens. Clinically, this is interpreted as a risk factor for CIPN at high cumulative doses of taxanes, warranting the closer monitoring and consideration of dose modifications. Data on *CYP2C8* rs1058930 remain limited and inconsistent [31].

### 4.4. CYP3A5

The *CYP3A5* rs776746 variant has been investigated as a potential modifier of irinotecan- and taxane-related toxicity due to altered drug clearance. However, results have been inconclusive. Several cohort studies have shown that associations with clinically relevant CIPN and neutropenia are not reproducible after adjustment for non-drug risk factors and concomitant medications [32]. To date, *CYP3A5* remains an area of investigation in oncology, with no consistent clinical recommendations available.

### 4.5. GSTP1

The frequencies of *GSTP1* rs1695 in this study were comparable to those reported in the European population. Several studies of colorectal cancer patients treated with FOLFOX have described an association between the rs1695 variant and cumulative oxaliplatin-induced neuropathy [33]. However, the direction and magnitude of the effect vary across studies, likely due to differences in dosage, treatment schedules, and criteria for assessing neurotoxicity. The rs1695 variant is currently considered an appropriate modifier of CIPN risk for inclusion in multigene panels but remains insufficient for independent clinical decision-making [34].

### 4.6. ERCC1

The *ERCC1* gene (rs11615 and rs3212986) has been implicated in platinum adduct repair. Meta-analyses and aggregated reviews of platinum-treated cohorts, including colorectal cancer, gastrointestinal tumors, and non–small-cell lung cancer (NSCLC), have reported associations between rs11615 and both treatment response and overall survival [35]. However, the most consistent results have been reported in Asian populations, whereas studies in European cohorts have often yielded inconclusive findings. In clinical practice, single *ERCC1* markers are not recommended as independent predictors of oxaliplatin efficacy or toxicity but may be included in polygenic panels.

### 4.7. XPC

The *XPC* rs2228001 polymorphism has been more extensively investigated in relation to colorectal cancer risk, whereas data on its role in predicting response or toxicity to platinum-based regimens remain limited and contradictory. Meta-analyses of platinum-based regimens have shown the limited diagnostic value of *XPC* variants for predicting treatment response, with no consistent associations identified for CIPN or hepatotoxicity [36].

### 4.8. CDA

The enzyme cytidine deaminase (CDA) inactivates gemcitabine, and genetic variation can substantially affect therapy tolerability. The rs2072671 polymorphism is associated with reduced enzyme activity, resulting in slower drug clearance and an increased risk of hematologic toxicity. Studies in pancreatic cancer patients have shown that carriers of the C allele are at higher risk of neutropenia and thrombocytopenia, a finding supported by pharmacokinetic data demonstrating increased exposure to gemcitabine [37,38].

In addition, some reports suggest a possible association between the C allele and increased treatment efficacy in gastrointestinal tumors, including pancreatic cancer and cholangiocarcinoma, likely due to increased exposure to the active drug. However, study results remain contradictory, and the impact of rs2072671 on survival is still uncertain. Therefore, this polymorphism may be considered an additional marker of toxicity risk in gemcitabine-based treatment of gastrointestinal tumors, but further validation in prospective clinical studies is required.

### 4.9. SLC31A1 (CTR1)

*SLC31A1* encodes CTR1, a copper transporter responsible for the uptake of copper ions and platinum compounds into cells. Experimental studies have demonstrated a direct correlation between CTR1 expression and the intracellular accumulation of cisplatin and oxaliplatin. Furthermore, reduced CTR1 expression has been associated with the development of drug resistance.

The most detailed studies have been conducted in patients with non–small-cell lung cancer, where the rs2233914 polymorphism has been associated with changes in both overall survival and the risk of toxicity during platinum-based treatment [39]. Direct studies are much less common in patients with gastrointestinal tumors, including colorectal cancer. Nevertheless, the involvement of CTR1 in response to platinum-based regimens (e.g., FOLFOX) is biologically plausible, as drug uptake and intracellular accumulation are critical steps in its mechanism of action. In this context, *SLC31A1* appears to be a promising marker for inclusion in expanded pharmacogenetic panels when assessing both toxicity risk and the efficacy of oxaliplatin therapy in gastrointestinal tumors. However, its clinical utility still requires confirmation in large prospective studies.

### 4.10. MTHFR

The contribution of the *MTHFR* rs1801133 polymorphism to fluoropyrimidine toxicity and efficacy remains under debate. In cohorts of Asian and Latin American colorectal cancer patients treated with 5-fluorouracil (5-FU) or capecitabine, carriage of the T allele particularly in the homozygous TT genotype has repeatedly been associated with an increased incidence of hematologic and gastrointestinal toxicity [40]. This decrease in MTHFR enzyme activity alters folate metabolism, thereby enhancing the toxic effects of fluoropyrimidines.

However, large-scale studies in European populations have often yielded neutral results, suggesting that this polymorphism has no relevant impact on therapy effectiveness or the incidence of serious adverse events [41]. This contrast in results may be explained by ethnic differences in allele distribution, variations in treatment regimens (monotherapy versus combination therapy), and differences in study design. In conclusion, the *MTHFR* rs1801133 polymorphism should be regarded as an additional marker with potential prognostic value in certain ethnic groups but is not a clinically validated biomarker for routine practice.

### 4.11. TYMS

The *TYMS* 3′UTR 6 bp ins/del (rs11280056) and 5′UTR VNTR variants have been extensively investigated for their role in modulating fluoropyrimidine response and toxicity. Separate studies of pancreatic and colorectal cancer patients treated with capecitabine found that rs11280056 was associated with an increased risk of hand–foot syndrome [42]. However, other studies have reported neutral or even protective effects, underscoring the heterogeneity of the findings.

Our study confirms the relevance of certain pharmacogenetic markers (*DPYD* and *UGT1A1*) as clinically validated biomarkers of chemotherapy toxicity. The role of other variants (*GSTP1, ERCC1, MTHFR, TYMS, SLC31A1*, and *CDA*) remains uncertain and requires further validation. The ethnic differences in allele frequencies identified in this cohort underscore the importance of conducting region-specific pharmacogenetic studies and implementing a personalized approach to the selection and dosing of cytotoxic drugs in clinical practice. Our frequency estimates largely mirror European data, consistent with cohort ancestry. Notable deviations included a higher prevalence of *UGT1A1*28* and a modest enrichment of *DPYD* rs2297595. While within published ranges, these differences may have clinical relevance (irinotecan and fluoropyrimidines, respectively) and warrant validation in larger, multi-center Russian cohorts with broader ethnic representation.

Limitations of this study include its single-center design, lack of detailed ethnic stratification, and the exploratory nature of genotype–phenotype assessment. Future multi-center studies with expanded geographic and ethnic coverage, combined with clinical outcome data, are needed to confirm the observed trends and support the development of population-specific pharmacogenetic recommendations for gastrointestinal cancers in Russia. The study cohort largely represented patients of European ancestry from the Central region of Russia, consistent with the allele frequencies observed. Future multi-center studies including other ethnic groups across Russia are warranted to evaluate potential inter-ethnic variability and enhance the representativeness of pharmacogenetic reference data.

## 5. Conclusions

A prospective study of 412 patients with gastrointestinal tumors was conducted to analyze key pharmacogenetic polymorphisms influencing the efficacy and toxicity of anticancer chemotherapy. The frequencies of most variants corresponded to those reported in the European population, but notable differences were observed for several genes. *DPYD* and *UGT1A1* remain the most clinically relevant variants, for which international dose-adjustment recommendations already exist. Other polymorphisms, such as *CYP2C8, GSTP1, ERCC1, MTHFR, TYMS*, and *SLC31A1*, may serve as markers for the development of expanded pharmacogenetic panels. The findings highlight the clinical utility of pre-treatment pharmacogenetic testing, especially for *DPYD* and *UGT1A1* variants, to individualize fluoropyrimidine- and irinotecan-based chemotherapy in Russian patients. The incorporation of these assays into standard oncology protocols could reduce toxicity, improve treatment safety, and form the basis for population-adapted pharmacogenetic programs in Russia.

## Figures and Tables

**Figure 1 genes-16-01261-f001:**
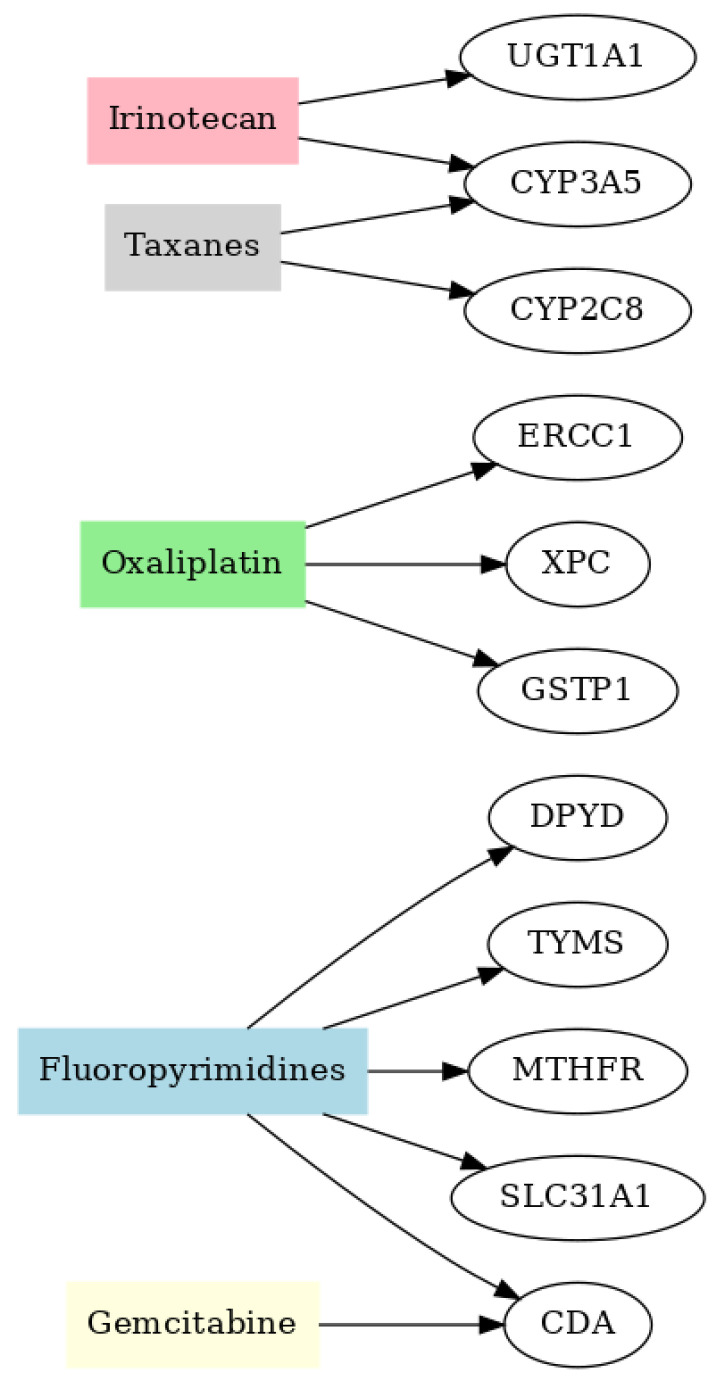
Flowchart of anticancer drugs used in the study and the pharmacogenes associated with their metabolism and toxicity.

**Figure 2 genes-16-01261-f002:**
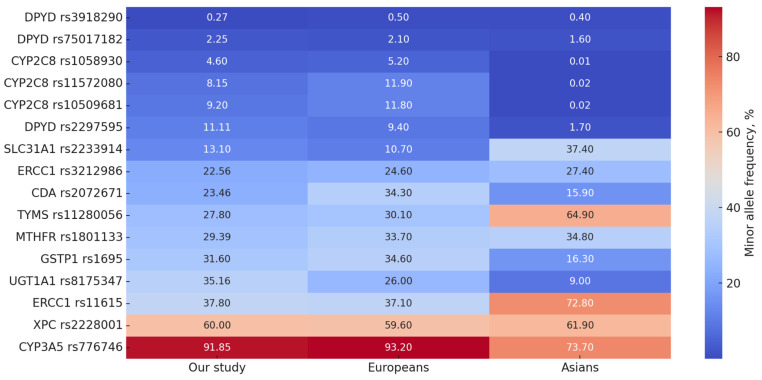
Heatmap of minor allele frequencies, sorted from lowest to highest in the study cohort.

**Table 1 genes-16-01261-t001:** Genotype distribution of pharmacogenetic variants in the study cohort.

Gene	Genotypes		χ2	*p*	Minor Allele Frequency (%)	Minor Allele Frequency in the European Population (%)	Minor Allele Frequency in the Asian Population (%)
*DPYD* rs2297595 [7]	TT	274	3.245	0.071	11.11%	9.4%	1.7%
	TC	76					
	CC	1					
*DPYD* rs3918290 [8]	GG	363	0.003	0.958	0.27%	0.5%	0.4%
	GA	2					
	AA	0					
*DPYD* rs75017182 [9]	GG	383	3.287	0.069	2.25%	2.1%	1.6%
	GC	16					
	CC	1					
*ERCC1* rs3212986 [10]	GG	241	1.894	0.169	22.56%	24.6%	27.4%
	GT	153					
	TT	16					
*ERCC1* rs11615 [11]	AA	152	1.920	0.166	37.8%	37.1%	72.8%
	AG	206					
	GG	52					
*GSTP1* rs1695 [12]	AA	190	0.016	0.900	31.6%	34.6%	16.3%
	AG	174					
	GG	41					
*UGT1A1* rs8175347 [13]	6\6	173	0.938	0.333	35.16%	26%	9%
	6\7	174					
	7\7	54					
*CYP3A5* rs776746 [14]	AA	2	0.209	0.647	91.85%	93.2%	73.7%
	AG	62					
	GG	341					
*CYP2C8* rs10509681 [15]	TT	335	0.816	0.366	9.2%	11.8%	0.02%
	CT	65					
	CC	5					
*CYP2C8* rs11572080 [16]	CC	339	3.187	0.074	8.15%	11.9%	0.02%
	CT	66					
	TT	0					
*CY2C8* rs1058930 [17]	GG	370	1.733	0.188	4.6%	5.2%	0.008%
	CG	33					
	CC	2					
*SLC31A1* rs2233914 [18]	GG	305	0.167	0.683	13.1%	10.7%	37.4%
	AG	94					
	AA	6					
*CDA* rs2072671 [19]	AA	241	1.058	0.304	23.46%	34.3%	15.9%
	AC	138					
	CC	26					
*XPC* rs2228001 [20]	CC	72	1.734	0.188	60%	59.6%	61.9%
	AC	184					
	AA	154					
*MTHFR* rs1801133 [21]	CC	204	0.009	0.921	29.39%	33.7%	34.8%
	CT	171					
	TT	35					
*TYMS* rs11280056 [22]	non-del	214	0.006	0.941	27.8%	30.1%	64.9%
	ndel/del	164					
	del	32					

## Data Availability

The datasets generated and analyzed during this study are not publicly available due to ethical restrictions and patient confidentiality protections under Russian Federation laws on personal data protection (Federal Law No. 152-FZ). However, anonymized data supporting the findings may be made available upon reasonable request from qualified researchers, subject to approval by the Local Ethics Committee of the Russian Medical Academy of Continuous Professional Education (contact: rmapo@rmapo.ru). Requests should include a detailed research proposal and data protection plan.

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
