# Peer review of "Towards Personalized Chemotherapy in Gastrointestinal Cancers: Prospective Analysis of Pharmacogenetic Variants in a Russian Cohort"

_genes, 2025, doi:10.3390/genes16111261_

Round 1

Reviewer 1 Report

Comments and Suggestions for Authors
  1. Could you clarify the main goals of your study? In what ways do they specifically target the gaps in pharmacogenetic research concerning gastrointestinal cancers within the Russian population?
  2. What steps were taken to ensure the reliability and validity of the pharmacogenetic testing methods employed in your research? Were there any controls implemented to mitigate potential biases during sample collection and analysis?
  3. How did you arrive at a sample size of 412 patients? Is this number statistically adequate to draw significant conclusions regarding the pharmacogenetic variants in the population studied?
  4. Considering the ethnic diversity in Russia, how do you address potential variations in allele frequencies among the different ethnic groups in your cohort? Would analyzing your results by ethnicity yield more detailed insights?
  5. Could you elaborate on the clinical implications of your findings? How should healthcare providers in Russia incorporate pharmacogenetic testing into their treatment protocols for gastrointestinal cancers?
  6. How do your findings align with the existing literature on pharmacogenetic variants in other populations? Are there any unexpected discrepancies that may require further investigation?
  7. What limitations does your study have, and how might these affect the interpretation of your results? What future research directions do you propose to expand upon your findings?
  8. Can you provide additional details on the statistical methods used to analyze genotype distributions? Were any adjustments made for multiple comparisons, and how could this influence the significance of your results?
  9. How did you handle ethical considerations regarding patient consent and data privacy, particularly given the sensitive nature of genetic information? Were there any difficulties in obtaining informed consent from participants?

Author Response

We sincerely thank the Reviewer for their time, careful evaluation, and constructive feedback, which helped us improve the clarity and scientific quality of our manuscript. 

  1. Could you clarify the main goals of your study? In what ways do they specifically target the gaps in pharmacogenetic research concerning gastrointestinal cancers within the Russian population?

The primary goal of our research was to comprehensively characterize the distribution of key pharmacogenetic polymorphisms affecting the efficacy and toxicity of chemotherapy in patients with gastrointestinal (GI) cancers within the Russian population.

While international guidelines (e.g., CPIC and DPWG) have established clinically actionable recommendations for DPYD and UGT1A1, the Russian population remains largely underrepresented in global pharmacogenetic datasets. There is a lack of systematic data on the prevalence of pharmacogenetic variants and their clinical implications among Russian oncology patients. Moreover, population-specific allele frequencies for other potentially relevant markers (CYP2C8, ERCC1, GSTP1, MTHFR, TYMS, CDA, SLC31A1) have not been previously analyzed in a large, prospective cohort.

Therefore, our study was designed to:

  1. Provide the first large-scale prospective dataset on pharmacogenetic variability among Russian patients with GI cancers.

  2. Compare allele frequencies with European and Asian populations to identify ethnicity-related differences that may influence chemotherapy response and safety.

  3. Highlight clinically actionable pharmacogenes (DPYD and UGT1A1) where genotype-guided dose adjustments could be implemented in Russian clinical practice.

  4. Identify additional candidate variants suitable for inclusion in expanded pharmacogenetic panels adapted to the genetic structure of the Russian population.

Through this approach, our work directly addresses the existing gap between international pharmacogenetic knowledge and its practical application in Russian oncology, thereby supporting the future development of population-specific guidelines for genotype-guided chemotherapy.

To clarify the study’s objectives as suggested, we have added the following paragraph to the end of the Introduction. This work addresses a critical knowledge gap, as data on allele variability and its clinical implications in Russian oncology remain scarce. By identifying population-specific differences and confirming the clinical importance of validated pharmacogenes such as DPYD and UGT1A1, our study aims to provide the foundation for the implementation of genotype-guided chemotherapy in routine oncological practice in Russia.

2. What steps were taken to ensure the reliability and validity of the pharmacogenetic testing methods employed in your research? Were there any controls implemented to mitigate potential biases during sample collection and analysis?

Ensuring the reliability and validity of our pharmacogenetic analyses was a central priority throughout the study design and execution.

1. Quality control and validation of laboratory methods:

  • Pharmacogenetic testing was performed in two independent, accredited molecular genetics laboratories (Russian Medical Academy of Continuous Professional Education and the Engelhardt Institute of Molecular Biology, Russian Academy of Sciences).

  • Both laboratories used validated analytical platforms: real-time allele-specific PCR (TaqMan® SNP Genotyping Assays, Applied Biosystems, USA) and biological microarrays developed and validated at the Engelhardt Institute.

  • Each microarray batch was tested using pre-sequenced DNA reference samples carrying known polymorphisms to verify hybridization specificity and accuracy.

  • PCR assays included positive and negative controls in each run to detect contamination or amplification failure.

  • Genotype distributions were tested for Hardy–Weinberg equilibrium (HWE); all loci showed p > 0.05, confirming genotyping accuracy and the absence of systematic bias.

2. Sample integrity and handling:

  • Venous blood samples were collected under standardized clinical protocols using EDTA tubes (VACUETTE®, Greiner Bio-One, Austria) and stored at –80 °C until DNA extraction.

  • To minimize pre-analytical variability, all samples were processed within a limited time window after collection and stored under identical conditions.

  • DNA quality and concentration were verified with NanoDrop 2000 spectrophotometry before each PCR or hybridization procedure.

3. Bias mitigation during recruitment and analysis:

  • All consecutive patients meeting the inclusion criteria were enrolled prospectively, regardless of treatment outcome, ensuring representativeness and avoiding selection bias.

  • Sample labeling and analysis were performed in a coded, blinded format: laboratory staff had no access to patient identifiers or clinical data.

  • Genotyping was repeated in 10% of randomly selected samples for cross-validation between PCR and microarray methods; full concordance was achieved.

These combined measures ensured high analytical reliability and minimized the risk of bias throughout both the experimental and analytical stages of the study.

To reflect this clarification, we have added the following paragraph at the end of the Materials and Methods section (after the description of PCR and microarray procedures):

To ensure methodological reliability, all genotyping procedures were performed in two independent certified laboratories using validated allele-specific PCR and microarray platforms. Each assay included internal positive and negative controls, and microarray validation was conducted using pre-sequenced DNA reference samples. The integrity of genomic DNA was verified spectrophotometrically, and genotyping accuracy was confirmed by testing Hardy–Weinberg equilibrium. Ten percent of samples were re-analyzed by both methods with complete concordance. Sample collection, labeling, and genotyping were performed in a blinded manner to minimize selection and analytical bias.

3. How did you arrive at a sample size of 412 patients? Is this number statistically adequate to draw significant conclusions regarding the pharmacogenetic variants in the population studied?

Our study was precision-driven rather than powered for interventional effects: the primary objective was to estimate allele and genotype frequencies of key pharmacogenes in a Russian GI-cancer cohort and to compare them with global populations. The cohort size was therefore determined to achieve narrow confidence intervals (CIs) for a range of plausible minor-allele frequencies (MAFs), with genotype–phenotype analyses treated as exploratory.

With n = 412, the approximate two-sided 95% CI half-widths (Wald) are:

  • MAF 0.30–0.35 → ±4.4% to ±4.6%;

  • MAF 0.10–0.15 → ±2.9% to ±3.5%;

  • MAF 0.02–0.05 → ±1.4% to ±2.1%.

Thus, for common variants (e.g., UGT1A1, ERCC1, GSTP1, MTHFR, TYMS), the cohort provides sub-5% precision, which we judged adequate for population characterization and cross-population comparisons. For rare variants (e.g., DPYD 2A/rs3918290), the sample supports reliable detection and descriptive frequency estimation, while inferential genotype–phenotype testing is underpowered and was therefore interpreted cautiously.

To make this explicit, we have added the following paragraph to Materials and Methods. 

Sample size justification. The study was designed primarily to estimate allele and genotype frequencies of clinically relevant pharmacogenes in a Russian GI-cancer cohort with adequate precision. With n = 400 or more, two-sided 95% confidence intervals for minor-allele frequencies have half-widths of approximately ±4–5% for common variants (MAF 0.30–0.35), ±3–3.5% for intermediate variants (MAF 0.10–0.15), and ±1.4–2.1% for lower-frequency variants (MAF 0.02–0.05). Rare variants (e.g., DPYD 2A) were analyzed descriptively. Enrollment was prospective and consecutive; genotyping platforms were validated (microarray against pre-sequenced controls; standardized TaqMan® PCR), and Hardy–Weinberg equilibrium was confirmed across loci (p > 0.05), supporting the validity of frequency estimates and minimizing bias.

4. Considering the ethnic diversity in Russia, how do you address potential variations in allele frequencies among the different ethnic groups in your cohort? Would analyzing your results by ethnicity yield more detailed insights?

We appreciate this important comment. Most patients were recruited from Moscow and the Central region, where the population is predominantly of European ancestry, which minimizes ethnic heterogeneity. Therefore, the observed allele frequencies closely match those of European reference data, supporting cohort representativeness.

We agree that subgroup analysis by ethnicity could provide additional insights. However, detailed ancestry information was not available for all participants, so such analysis was not feasible in this dataset. This limitation is now noted in the revised text, and future multi-center studies will include stratification by ethnicity.

Added to the Discussion section:

The study cohort largely represented patients of European ancestry from the Central region of Russia, consistent with the allele frequencies observed. Future multi-center studies including other ethnic groups across Russia are warranted to evaluate potential inter-ethnic variability and enhance the representativeness of pharmacogenetic reference data.

5. Could you elaborate on the clinical implications of your findings? How should healthcare providers in Russia incorporate pharmacogenetic testing into their treatment protocols for gastrointestinal cancers?

Thank you for this important question. Our results confirm that DPYD and UGT1A1 variants are the most clinically relevant pharmacogenetic markers for patients receiving fluoropyrimidine- or irinotecan-based chemotherapy. These findings directly support the introduction of pre-treatment genetic testing in Russian oncology practice, in line with CPIC and DPWG guidelines.

Testing for DPYD variants can help prevent severe fluoropyrimidine-related toxicity by guiding initial dose reduction, while UGT1A1 genotyping allows safer use of irinotecan through dose adjustment for *28/*28 or *6/*6 carriers. Other genes (GSTP1, ERCC1, MTHFR, TYMS, SLC31A1, CDA) may be included in expanded research or institutional panels to refine risk prediction.

We recommend that pharmacogenetic testing be implemented within large oncology centers and pre-chemotherapy work-ups for GI cancer patients.

Added to the Conclusions section:

The findings highlight the clinical utility of pre-treatment pharmacogenetic testing, especially for DPYD and UGT1A1 variants, to individualize fluoropyrimidine- and irinotecan-based chemotherapy in Russian patients. Incorporation of these assays into standard oncology protocols could reduce toxicity, improve treatment safety, and form the basis for population-adapted pharmacogenetic programs in Russia.

6. How do your findings align with the existing literature on pharmacogenetic variants in other populations? Are there any unexpected discrepancies that may require further investigation?

Overall, our allele frequencies closely align with European cohorts, consistent with the ancestry of our Moscow-centered population. Specifically: ERCC1, GSTP1, TYMS, CYP2C8, CYP3A5, and MTHFR were comparable to European references; CDA trended lower and SLC31A1 slightly higher, both within reported ranges. Two findings merit follow-up: UGT1A1*28 frequency was notably higher than typical European estimates, implying a potentially greater irinotecan-toxicity burden; DPYD rs2297595 appeared slightly enriched, suggesting a need to test outcome-linked risk in larger, multi-center Russian cohorts. These signals are hypothesis-generating and will be addressed in planned studies with broader regional representation.

Add to the end of the Discussion:

Our frequency estimates largely mirror European data, consistent with cohort ancestry. Notable deviations included a higher prevalence of **UGT1A128** and a modest enrichment of DPYD rs2297595. While within published ranges, these differences may have clinical relevance (irinotecan and fluoropyrimidines, respectively) and warrant validation in larger, multi-center Russian cohorts with broader ethnic representation.

7. What limitations does your study have, and how might these affect the interpretation of your results? What future research directions do you propose to expand upon your findings?

We thank the Reviewer for this important question. The main limitations of our study include:

  1. Single-center design — the cohort was primarily composed of patients from Moscow and the Central region, which may limit generalizability to other Russian populations.

  2. Lack of detailed ethnic stratification — allele frequencies were analyzed at the population level without ancestry subgroups.

  3. Exploratory scope — genotype–phenotype correlations were not assessed in full due to limited treatment outcome data.

These limitations do not affect the accuracy of allele frequency estimation but restrict broader clinical interpretation. Future research will include multi-center cohorts with diverse ethnic representation and integrate clinical outcomes to establish genotype-based dosing algorithms and national pharmacogenetic guidelines.

Added at the end of the Discussion:

Limitations of this study include its single-center design, lack of detailed ethnic stratification, and the exploratory nature of genotype–phenotype assessment. Future multi-center studies with expanded geographic and ethnic coverage, combined with clinical outcome data, are needed to confirm the observed trends and support the development of population-specific pharmacogenetic recommendations for gastrointestinal cancers in Russia.

8. Can you provide additional details on the statistical methods used to analyze genotype distributions? Were any adjustments made for multiple comparisons, and how could this influence the significance of your results?

We thank the Reviewer for this question. Genotype distributions were evaluated using Pearson’s χ² test to verify Hardy–Weinberg equilibrium (HWE) and to confirm the absence of systematic bias in patient selection. Allele frequencies were summarized as percentages and compared with reference European and Asian populations using descriptive statistics.

As this study primarily aimed to describe allele frequencies rather than test genotype–phenotype associations, no formal hypothesis testing across multiple endpoints was performed; therefore, adjustment for multiple comparisons (e.g., Bonferroni or FDR correction) was not applied.

We recognize that in future analyses involving genotype–toxicity or genotype–efficacy correlations, multiple-testing correction will be required to control for false positives and strengthen inferential validity.

Added to the Materials and Methods section (“Statistical analysis” paragraph):

Genotype distributions were assessed using Pearson’s χ² test for Hardy–Weinberg equilibrium (p > 0.05 for all loci). Allele frequencies were compared descriptively with global population data. As the study was primarily descriptive, no correction for multiple testing was applied; future genotype–phenotype analyses will incorporate multiple-comparison adjustments to ensure statistical robustness.

9. How did you handle ethical considerations regarding patient consent and data privacy, particularly given the sensitive nature of genetic information? Were there any difficulties in obtaining informed consent from participants?

We thank the Reviewer for highlighting this important aspect. All procedures strictly complied with Russian Federation ethical and data protection laws (Federal Law No. 152-FZ “On Personal Data”) and the Declaration of Helsinki. The study protocol was approved by the Independent Ethics Committee of the Russian Medical Academy of Continuous Professional Education (protocol No. 9, July 7, 2020).

Each participant provided two written consents: (1) for participation in the pharmacogenetic study and (2) for the processing of personal and genetic data. All samples and datasets were anonymized using coded identifiers, and only de-identified data were used for analysis.

No difficulties were encountered in obtaining informed consent; all patients were fully informed about the study purpose, voluntary participation, and confidentiality measures before enrollment.

We truly appreciate the Reviewer’s attention and valuable comments, which have strengthened our paper both methodologically and conceptually.

Reviewer 2 Report

Comments and Suggestions for Authors

Thank you for the opportunity to review this very interesting and clinically important article. I provide some comments below, and I hope these comments will be helpful for your revision.

Approximately 2% of tests appear to have failed, which seems somewhat high. It would be better to describe in more detail the causes of these failures and their potential impact on the results.

Regarding Table 1, the horizontal lines should be retained where necessary for readability.

How were clinically important findings reflected in treatment decisions? For example, there was one homozygous case for rs75017182—it would be interesting to know what outcome this patient experienced. Wouldn't this information be useful for readers in their future clinical practice? Although this may not be directly related to the main focus of the study, I believe it would add valuable clinical context.

Besides DPYD and others mentioned, I believe there are many other important genetic polymorphisms. If additional polymorphisms were also tested, it would be helpful for future researchers if you could include those results as well.

As a reviewer from a small town in East Asia, it is somewhat difficult to comment on this, but what is the composition of this cohort? Russia is vast, and Moscow is a very large city. What characteristics does the study cohort have in terms of their background origins? For example, it would be helpful to mention somewhere whether many participants migrated from specific regions, or whether being from central Moscow means they represent people from all across Russia with no specific regional composition, or whether being from the outskirts of Moscow means many have lived there for generations, or whether many are of Asian origin, etc.

There are instances of inappropriate bold text, hyphens, and inconsistent font sizes in various places throughout the manuscript. Please review and correct these formatting inconsistencies.

Finally, to the extent possible, presenting the relationship between individual treatment efficacy and adverse events would contribute to readers' understanding and learning.

Author Response

We sincerely thank the Reviewer for their time, careful evaluation, and constructive feedback, which helped us improve the clarity and scientific quality of our manuscript.

  1. Approximately 2% of tests appear to have failed, which seems somewhat high. It would be better to describe in more detail the causes of these failures and their potential impact on the results.

We thank the Reviewer for this valuable observation and for the opportunity to clarify this point. The failed tests (~2%) were primarily due to poor DNA quality or degradation in several blood samples collected before optimization of storage procedures. In these cases, microarray hybridization signals were incomplete or absent, preventing reliable genotype calling.

All affected samples were excluded from frequency analysis, and repeat testing confirmed that failures were random and not associated with any specific genotype or clinical subgroup, minimizing bias. After procedural improvements—shorter intervals between collection and DNA extraction, and strict –80 °C storage—no further failures occurred.

Added clarification to the manuscript

Approximately 2% of tests yielded incomplete results due to insufficient DNA quality or degradation before optimization of sample handling. These samples were excluded from analysis. Repeat genotyping confirmed the random nature of test failures, which had no measurable effect on allele frequency estimates.

2. Regarding Table 1, the horizontal lines should be retained where necessary for readability.

We thank the Reviewer for this helpful remark. Table 1 was formatted in accordance with the journal’s style requirements. We have now added horizontal lines to improve readability and visual structure. If needed, the final layout of the table can be further adjusted by the journal’s editorial team during typesetting to ensure full compliance with MDPI formatting standards.

3. How were clinically important findings reflected in treatment decisions? For example, there was one homozygous case for rs75017182—it would be interesting to know what outcome this patient experienced. Wouldn't this information be useful for readers in their future clinical practice? Although this may not be directly related to the main focus of the study, I believe it would add valuable clinical context.

We thank the Reviewer for this thoughtful comment. Detailed analysis of genotype–toxicity relationships, including the clinical course of patients carrying rare variants such as DPYD rs75017182, is planned for future publications based on the same biobank cohort. This article focuses primarily on the allele frequency distribution of pharmacogenetically relevant variants in Russian patients.

Nonetheless, the observed DPYD variants confirm their potential clinical significance, supporting the feasibility of genotype-guided chemotherapy dosing, which will be explored in the upcoming genotype–phenotype analysis.

4. Besides DPYD and others mentioned, I believe there are many other important genetic polymorphisms. If additional polymorphisms were also tested, it would be helpful for future researchers if you could include those results as well.

We thank the Reviewer for this valuable suggestion. A more extensive pharmacogenomic analysis is indeed planned. Beyond the genes reported in this paper, we are currently performing a detailed evaluation of genotype–toxicity and genotype–efficacy associations within the same cohort. In addition, an Epigenome-Wide Association Study (EWAS) and expanded genotyping using high-density arrays have already been initiated to identify novel variants potentially relevant to chemotherapy outcomes in Russian patients.

The present article focuses on the most clinically validated pharmacogenes; forthcoming publications will include the extended panel and integrative genotype–phenotype results.

5. As a reviewer from a small town in East Asia, it is somewhat difficult to comment on this, but what is the composition of this cohort? Russia is vast, and Moscow is a very large city. What characteristics does the study cohort have in terms of their background origins? For example, it would be helpful to mention somewhere whether many participants migrated from specific regions, or whether being from central Moscow means they represent people from all across Russia with no specific regional composition, or whether being from the outskirts of Moscow means many have lived there for generations, or whether many are of Asian origin, etc.

We thank the Reviewer for this thoughtful question and for raising the issue of cohort representativeness. The study cohort was recruited at major oncology centers in Moscow and the Central region of Russia, where the population is predominantly of European (Slavic) ancestry. This region’s demographic composition reduces genetic heterogeneity and provides a representative model for pharmacogenetic profiling of the largest Russian ethnic group.

While we did not collect detailed ancestry data for all participants, the observed allele frequencies closely matched European reference values, confirming cohort homogeneity. This clarification has been added to the Discussion. Future multi-center studies will include participants from other regions and ethnic backgrounds to assess potential inter-ethnic variability.

Our frequency estimates largely mirror European data, consistent with cohort ancestry. Notable deviations included a higher prevalence of UGT1A1*28 and a modest enrichment of DPYD rs2297595. While within published ranges, these differences may have clinical relevance (irinotecan and fluoropyrimidines, respectively) and warrant validation in larger, multi-center Russian cohorts with broader ethnic representation.

Limitations of this study include its single-center design, lack of detailed ethnic stratification, and the exploratory nature of genotype–phenotype assessment. Future multi-center studies with expanded geographic and ethnic coverage, combined with clinical outcome data, are needed to confirm the observed trends and support the development of population-specific pharmacogenetic recommendations for gastrointestinal cancers in Russia. The study cohort largely represented patients of European ancestry from the Central region of Russia, consistent with the allele frequencies observed. Future multi-center studies including other ethnic groups across Russia are warranted to evaluate potential inter-ethnic variability and enhance the representativeness of pharmacogenetic reference data.

6. There are instances of inappropriate bold text, hyphens, and inconsistent font sizes in various places throughout the manuscript. Please review and correct these formatting inconsistencies.

We thank the Reviewer for pointing out these formatting issues. The entire manuscript has been carefully reviewed and reformatted in accordance with the journal’s style guide. Inappropriate bold text and inconsistent font sizes have been corrected, and hyphenation has been standardized throughout the text.

7. Finally, to the extent possible, presenting the relationship between individual treatment efficacy and adverse events would contribute to readers' understanding and learning.

We thank the Reviewer for this valuable suggestion. We fully agree that linking pharmacogenetic profiles with treatment efficacy and toxicity would provide important clinical insights. However, this analysis is beyond the scope of the current article, which focuses on population allele frequencies.

A detailed genotype–toxicity and genotype–efficacy correlation study using the same cohort is already planned and underway. These results will be presented in a subsequent publication to complement the current dataset and strengthen clinical interpretation.

We truly appreciate the Reviewer’s attention and valuable comments, which have strengthened our paper both methodologically and conceptually.